# SELF-SUPERVISED DENOISING FOR SINGLE VOLUME MR ANGIOGRAPHY VIA SLICE-AWARE DIFFUSION

## ABSTRACT

Recent advances in deep learning have significantly improved medical image de-
noising, particularly through supervised convolutional neural network–based ap-
proaches. However, these rely on large-scale paired noisy–clean datasets, which
limits their practical deployment in clinical settings where clean references are
rarely available. Self-supervised methods mitigate this issue but typically depend
on multi-volume data or temporal consistency, making them unsuitable for sin-
gle 3D volume data like magnetic resonance angiography (MRA). We propose a
novel self-supervised denoising framework for single noisy volume that leverages
spatial coherence across adjacent slices to construct training pairs without clean la-
bels or repeated scans. At its core is a conditional denoising diffusion model with
expectation-only sampling, enabling robust signal recovery. To further enhance
anatomical fidelity, we introduce a patch-wise adaptive post-processing module
that refines spatially localized features to better preserve anatomical accuracy.
Validated on 7T and 3T time-of-flight MRA datasets, our method significantly
improves vessel visibility while suppressing noise, offering a clinically practical
denoising approach tailored to real-world imaging workflows.

## 1 INTRODUCTION

High-resolution medical imaging modalities such as magnetic resonance imaging (MRI) and com-
puted tomography (CT) are critical for clinical diagnosis and treatment planning Van Geuns et al.
(1999); Liguori et al. (2015). However, acquiring high-resolution volumetric data often involves
trade-offs with scan time, increased radiation exposure, or hardware limitations, resulting in noisy
acquisitions Zhu et al. (2009); Costello et al. (2013). In clinical workflows, noisy 3D vol-
umes—often acquired as single-shot scans due to practical constraints—pose persistent challenges
for downstream analysis such as segmentation or abnormal detection Saladi & Amutha Prabha
(2017).

Conventional denoising techniques—ranging from spatial filtering Wink & Roerdink (2004); Buades
et al. (2005); Manjón et al. (2008) to deep supervised models Zhang et al. (2017); Chen et al. (2017);
Zhang et al. (2018)—struggle in this setting. Classical methods tend to oversmooth anatomical struc-
tures due to rigid noise assumptions, while supervised learning approaches require large-scale paired
clean-noisy datasets that are rarely available in clinical workflows. This is particularly problematic
in scenarios such as high-resolution MRI , where acquiring clean ground-truth scans is impractical
or even infeasible.

Self-supervised learning Batson & Royer (2019); Quan et al. (2020); Kang et al. (2024) has emerged
as a promising alternative by learning to denoise directly from noisy data, removing the reliance on
clean labels during training. Commonly adopted strategies in self-supervised frameworks, such
as masked-pixel prediction or blind-spot networks, aim to prevent identity mapping during training.
However, these approaches can blur or inadvertently suppress sparsely represented anatomical struc-
tures. In modalities like MRA, this often leads to the attenuation or loss of fine vascular features
such as small peripheral arteries, which occupy only a small portion of the volume but are crucial
for clinical interpretation.

Recently, most existing self-supervised denoising methods using generative models rely on assump-
tions such as temporal consistency, multi-view redundancy, or repeated acquisitions. These assump-
tions do not hold in routine clinical workflows involving single 3D volume imaging. As a result,

existing frameworks may struggle to generalize to single-volume clinical data, limiting their practical utility in real-world diagnostic scenarios.

To address these limitations, we propose **Di-Flow**, a self-supervised denoising framework designed specifically for a *single* noisy medical volume. Rather than relying on external redundancy, Di-Flow leverages structural consistency within a single volume via a slice-aware *multi-slice averaging* (MSA) strategy that constructs pseudo training pairs directly from the noisy input. Building on these pairs, a conditional diffusion model with *expectation-only* inference produces deterministic restorations without requiring clean labels. In addition, we introduce a *patch-wise adaptive post-processing* (PAP) module that enhances fine anatomical detail and suppresses residual structured noise. Across 7T and 3T TOF-MRA datasets, Di-Flow consistently improves vascular SNR/CNR and preserves peripheral vessel continuity compared with classical, self-supervised, and diffusion-based baselines, demonstrating robustness to acquisition settings and practical suitability for clinical workflows.

## 2 RELATED WORK

### 2.1 TRADITIONAL AND SUPERVISED DENOISING METHODS

Early medical image denoising relied on non-learning algorithms including Gaussian smoothing Wink & Roerdink (2004), wavelet shrinkage Ouahabi (2013), and non-local means filtering Buades et al. (2005); Manjón et al. (2008), which suppress noise by leveraging spatial redundancy in anatomical structures. Advanced methods like BM3D Dabov et al. (2007) and BM4D Maggioni et al. (2012) introduced collaborative filtering of matched patches in transform domains. These traditional approaches assume stationary noise and use fixed filtering rules, making them less effective for clinical images with spatially varying noise and often causing oversmoothing of diagnostically important details Kulathilake et al. (2022). Their lack of adaptability to heterogeneous noise and anatomy has led to the rise of learning-based methods, which offer superior preservation of diagnostically relevant structures Thakur et al. (2024).

The rise of deep learning enabled supervised convolutional neural networks (CNNs) to achieve superior performance through methods like DnCNN Zhang et al. (2017) with residual learning; RED-CNN Chen et al. (2017) which leverages a residual encoder-decoder architecture; and FFD-Net Zhang et al. (2018) for handling diverse noise levels efficiently. While these models effectively learn anatomical structures and noise patterns from paired data, clinical deployment remains challenging due to the requirement for large paired clean-noisy datasets, which are rarely available in medical settings and ethically problematic to acquire, as doing so typically requires repeated scans under controlled conditions—thereby increasing patient risk, scan time, and costs—and often introducing motion artifacts.

### 2.2 SELF-SUPERVISED DENOISING METHODS

To overcome the limitations of supervised approaches, recent studies have explored self-supervised denoising frameworks that do not require clean reference images. Noise2Noise Lehtinen et al. (2018) demonstrated that clean targets are unnecessary when training on pairs of noisy images with independent, zero-mean noise. Building on this insight, Noise2Void Krull et al. (2019), Noise2Self Batson & Royer (2019), and Self2Self Quan et al. (2020) enable training from single noisy images through blind-spot networks, J-invariant prediction, and dropout-based augmentation respectively. Multidimensional Self2Self Kang et al. (2024) extends this framework by leveraging redundancy in repeated multidimensional MRI acquisitions. Despite promising results in general MRI denoising, these methods struggle in challenging scenarios where noise and underlying signal distributions overlap, such as angiographic imaging with sparse vasculature and similar-intensity background structures Li et al. (2021). Extensions like Noisier2Noise Moran et al. (2020) and Neighbor2Neighbor Huang et al. (2021) aim to improve performance under severe noise conditions by refining pixel selection strategies or enforcing consistency between pairs of noisy images. Nevertheless, reliably preserving the continuity and topology of fine vascular anatomy remains difficult for these methods.

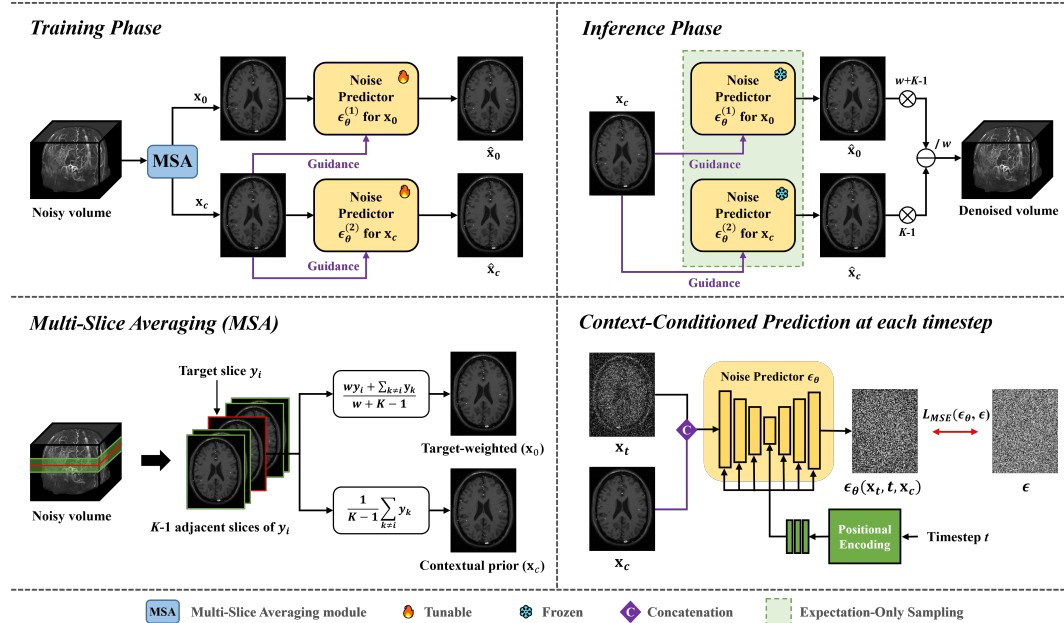

Figure 1: Overview of the proposed **Di-Flow** framework. The top panels illustrate the training and inference pipelines. During training, the model learns to denoise both the target-weighted input $x_0$ and the contextual prior $x_c$ using a dual-noise prediction scheme. During inference, only $x_c$ is required, and the final output is reconstructed through expectation-only sampling. The bottom-left panel describes the Multi-Slice Averaging (MSA) strategy, which constructs $x_0$ and $x_c$. The bottom-right panel shows the reverse denoising process at each timestep $t$, where the conditional diffusion model predicts noise $\epsilon$ from the noisy input $x_t$, timestep embedding, and contextual prior $x_c$.

## 2.3 GENERATIVE AND DIFFUSION-BASED METHODS

More recently, generative modeling approaches, particularly diffusion-based models, have shown significant promise in medical image denoising tasks due to their ability to model the full distribution of underlying clean images and iteratively refine noisy inputs. Methods such as DDM$^2$Xiang et al. (2023) and Di-FusionWu et al. (2025) have reported outstanding results, especially in diffusion-weighted MRI, by leveraging $J$-invariant property within temporal or multi-volume redundancy. However, these approaches typically depend on multiple acquisitions or repeated imaging, and therefore cannot be directly applied to common 3D imaging modalities. This motivates the need for novel diffusion-based methods specifically designed to leverage spatial coherence within single noisy volumes without external redundancy.

## 3 METHODOLOGY

We introduce Di-Flow, a self-supervised denoising framework tailored for single noisy MRA volumes. Our framework comprises three main components: (1) a Multi-Slice Averaging (MSA) module to create pseudo-training data, (2) conditional denoising diffusion models for robust signal restoration, and (3) a Patch-wise Adaptive Post-processing (PAP) module to refine local anatomical consistency. The following subsections provide a detailed description of each component.

### 3.1 MULTI-SLICE AVERAGING FOR DATA CONSTRUCTION

Given a noisy volume $Y = \{y_i\}_{i=1}^N$, where each slice $y_i \in \mathbb{R}^{H \times W}$ represents the $i$-th axial image, we construct pseudo-training pairs using the MSA module. The underlying assumption is that noise across slices is independent, thus statistically attenuated when averaging multiple scans, while coherent anatomical signals are preserved. Di-Flow exploits imaging characteristics by constructing two complementary inputs from $K$ adjacent slices of around the target slice $y_i$: The target-enhanced

pseudo-clean images $x_0$, emphasizing the target slice $y_i$; The contextual prior image $x_c$, representing averaged neighboring slices without the target.

$$x_0 = \frac{w y_i + \sum_{k \in K, k \neq i} y_k}{w + K - 1}, \quad x_c = \frac{1}{K - 1} \sum_{k \in K, k \neq i} y_k, \tag{1}$$

where the weighting factor $w$ controls the relative emphasis on the target slice $y_i$.

This averaging across slices significantly reduces effective noise power proportionally as the number of slices $K$ increases, directly enhancing the signal-to-noise ratio (SNR) of the pseudo-training pairs. By doing so, we can utilize anatomical continuity across adjacent slices to preserve anatomical structure while learning to suppress noise, especially for fine through-plane vasculature. Leveraging flow context from neighboring slices prevents subtle peripheral vessels, which are often obscured when denoising each slice in isolation, from being suppressed. By emulating paired data from within a single noisy volume, this design allows the model to operate in a self-supervised regime, as it does not rely on ground-truth images.

## 3.2 INTER-SLICE CONTEXTUAL DENOISING DIFFUSION MODEL

To recover clean target signals from the constructed pseudo pairs, we propose a denoising diffusion probabilistic framework based on DDPM Ho et al. (2020), with two coordinated branches. The primary branch reconstructs the target-enhanced image $x_0$ using a conditional diffusion model that incorporates the contextual prior $x_c$ at each step, enabling the model to utilize spatial context from adjacent slices during denoising. In parallel, an auxiliary self-reconstruction branch is trained to denoise the contextual prior $x_c$ using self-conditioning, where both the noisy input and the conditioning context are set to $x_c$. This approach allows the network to restore anatomical structures that are robustly preserved through spatial averaging of neighboring slices, while effectively suppressing inherent noise components.

For each timestep $t$, the forward diffusion process corrupts $x_0$ with Gaussian noise as in Eq. 2.

$$x_t = \sqrt{\bar{\alpha}_t}\, x_0 + \sqrt{1 - \bar{\alpha}_t}\, \epsilon, \qquad \epsilon \sim \mathcal{N}(0, \mathbf{I}), \tag{2}$$

where $\bar{\alpha}_t = \prod_{s=1}^{t} \alpha_s$ controls the cumulative noise strength. The neural network $\epsilon_\theta$, based on a U-Net architecture, predicts the noise given the noisy input $x_t$, the diffusion timestep $t$, and the conditioning context $x_c$. Training then minimizes the mean-squared error in Eq. 3.

$$\mathcal{L}_{\text{MSE}} = \mathbb{E}_{x_0, \epsilon, t}\left[ \|\epsilon - \epsilon_\theta(x_t, t, x_c)\|_2^2 \right] \tag{3}$$

This two-branch configuration enables the model to distinguish and recover the true underlying signal by simultaneously leveraging adjacent spatial context for denoising the target in the primary branch, while ensuring that only repeatable, shared information within the neighborhood is preserved in the auxiliary self-reconstruction branch. During the reverse diffusion process, the framework inherently suppresses stochastic noise while maintaining anatomically coherent features across slices. By optimizing both branches in parallel within a unified framework, the model is encouraged robust denoising for both target and contextual regions, without requiring any clean or external supervision.

### 3.2.1 INFERENCE VIA EXPECTATION-ONLY SAMPLING

In conventional DDPM inference, a stochastic sampling approach is typically adopted, $x_{t-1} \sim \mathcal{N}\big(\mu_\theta(x_t, t, x_c), \sigma_t^2 I\big)$, where randomness allows diverse outcomes suitable for generative tasks. However, for medical imaging applications, such variability can compromise the reliability necessary for clinical decision-making. To address this, we employ an expectation-only sampling strategy at inference. Rather than sampling from a distribution, we directly use the conditional mean prediction at each timestep (Eq. 4), which eliminates stochasticity and ensures stable preservation of anatomical details.

$$x_{t-1} = \mu_\theta(x_t, t, x_c) = \frac{1}{\sqrt{\alpha_t}}\left( x_t - \frac{1 - \alpha_t}{\sqrt{1 - \bar{\alpha}_t}}\, \epsilon_\theta(x_t, t, x_c) \right) \tag{4}$$

At inference, we generate two reconstructions: $\hat{x}_0$, a reconstruction including the target slice and its neighbors, and $\hat{x}_c$, the denoised contextual input. The final denoised target slice $\hat{y}_i$ is then recovered by analytically inverting the MSA averaging process according to Eq. 5. This explicit inversion disentangles the contribution of the contextual prior from the pseudo-target, allowing isolation of the denoised target slice.

$$\hat{y}_i = \frac{w + K - 1}{w}\hat{x}_0 - \frac{K - 1}{w}\hat{x}_c. \tag{5}$$

### 3.2.2 Relation to $J$-invariant.

This inversion results in a $J$-invariant reconstruction. The final procedure corresponds to the conditional expectation $\mathbb{E}[y_i|x_c]$, which is theoretically provided by the minimum mean squared error estimate under $J$-invariance assumptions, thereby enabling robust denoising without requiring clean supervision. (See Appendix A for details.)

### 3.3 Patch-Wise Adaptive Post-Processing

Despite the effectiveness of diffusion-based generative denoising methods, subtle anatomical inaccuracies or generated residual artifacts can persist, especially in vessel-rich modalities such as angiography. In these cases, small errors in reconstructing fine vascular structures can undermine clinical interpretability. To address this, we introduce a Patch-Wise Adaptive Post-processing (PAP) module that locally refines the denoised output by correcting voxel intensities based on patch-level statistical characteristics.

**Calculation of Difference Map.** The PAP module begins by identifying residual errors that remain after initial denoising. To this end, we compute a voxel-wise difference map $D = y_i - \hat{y}_i$, where $y_i$ is the original noisy slice and $\hat{y}_i$ is its denoised counterpart. Ideally, $D$ should resemble a noise distribution with a mean near zero if denoising has preserved anatomical structures accurately. However, in practice, localized deviation near vascular regions often signal structural degradation or artifacts, underscoring the need for targeted correction.

**Local Patch Classification.** Different anatomical regions require different correction strategies. To capture local characteristics, each slice is divided into overlapping patches $\{P_v\}$. For each patch centered at voxel $v$, we compute the local mean $\mu_v$ and standard deviation $\sigma_v$:

$$\mu_v = \frac{1}{|P_v|}\sum_{j \in P_v} y_j, \quad \sigma_v = \sqrt{\frac{1}{|P_v|}\sum_{j \in P_v}(y_j - \mu_v)^2}. \tag{6}$$

Using these statistics, patches are categorized as vessel-dominant (high intensity and high variability), background-dominant (low intensity and low variability), or normal regions (high intensity and low variability). This classification enables region-specific correction.

**Adaptive Histogram-Based Correction.** To selectively restore vessel signals while suppressing noise, we apply histogram-based clipping to the difference map $D$ using adaptive thresholds $[t_{\text{low}}, t_{\text{high}}]$ for each patch class. These thresholds are determined as confidence intervals of the difference histogram within each patch, reflecting the expected distribution of noise as well as potential signal loss or artifact emergence:

$$D_v^{\text{corrected}} = \begin{cases} D_v, & \text{if } t_{\text{low}} \leq D_v \leq t_{\text{high}}, \\ t_{\text{low}}, & \text{if } D_v < t_{\text{low}}, \\ t_{\text{high}}, & \text{if } D_v > t_{\text{high}}. \end{cases} \tag{7}$$

Vessel-dominant patches use narrower thresholds to recover small vascular signals, while background-dominant patches apply wider thresholds to suppress noise. By applying this correction strategy, the PAP module enables targeted refinement: true vascular structures can be restored if lost during denoising, while aggressive noise suppression is achieved in non-vascular regions. The final output after correction exhibits enhanced vessel fidelity and reduced spurious artifacts.

The corrected difference $D_v^{\text{corrected}}$ is subtracted from the original noisy slice $y_i$ to obtain the final refined output:

$$\hat{y}_i^{\text{refined}} = y_i - D_v^{\text{corrected}} \tag{8}$$

This operation effectively removes only the noise components identified through adaptive clipping, while preserving or recovering true vessel signals that may have been lost during denoising. Patch-wise correction is sequentially applied along the axial, sagittal, and coronal planes, yielding the final output after all three passes to ensure isotropic anatomical accuracy.

## 4    EXPERIMENTS

We evaluate the denoising performance of Di-Flow in comparison with state-of-the-art baseline methods. The baselines include both slice-wise and volumetric denoising methods, encompassing a traditional algorithm (BM3D Dabov et al. (2007)), self-supervised approaches (Noisier2Noise Moran et al. (2020), Neighbor2Neighbor Huang et al. (2021)), and diffusion-based self-supervised methods (DDM$^2$Xiang et al. (2023), Di-FusionWu et al. (2025)). We evaluated our method on clinically relevant high-resolution high-resolution time-of-flight (TOF) MRA datasets acquired at different field strengths.

### 4.1    EXPERIMENTAL SETUP

Experiments were conducted on high-resolution TOF-MRA datasets acquired at 7 Tesla (7T, MAGNETOM Terra) and 3 Tesla (3T, MAGNETOM Vida) scanners (Siemens Healthineers, Erlangen, Germany) from healthy volunteers. For both the 7T and 3T datasets, images were obtained from 5 subjects for training and 2 subjects for validation. The 7T TOF-MRA dataset comprises images with a spatial resolution of $800 \times 624$ pixels, resulting in 1,260 training slices and 504 validation slices. Similarly, the 3T TOF-MRA dataset contains images at a resolution of $450 \times 576$ pixels, with the same number of training and validation slices. Notably, all experiments were performed using only single-acquisition noisy volumes to emulate realistic clinical scenarios, where clean ground-truth references are unfeasible.

### 4.2    IMPLEMENTATION DETAILS

**Di-Flow Configuration.**    Pseudo-training pairs $(x_0, x_c)$ were generated using the proposed MSA strategy, combining $K - 1$ adjacent slices with a target slice weighted by $w$. Specifically, parameters were determined as $K = 5$, $w = 6$ for the 7T dataset, and $K = 9$, $w = 6$ for the 3T dataset, respectively, based on slice thickness and dataset-specific SNR characteristics (see Appendix B). The denoising framework was implemented as a conditional DDPM with a U-Net backbone comprising four resolution levels and channel multipliers of $(1, 2, 4, 8)$. At every diffusion step, noisy input $x_t$ and contextual prior $x_c$ were concatenated along the channel dimension and jointly fed into the network. The model was trained using the Adam optimizer with a learning rate of $8 \times 10^{-5}$, a batch size of 12, and a sampling timestep of 1000. For data augmentation, we applied random cropping to $256 \times 256$ and horizontal flipping. Expectation-only sampling was applied during inference. All experiments were implemented in PyTorch 2.4.1 with Python 3.8.5 and conducted on NVIDIA GeForce RTX 3090 GPUs with 24 GB memory.

**Competing Methods.**    All methods were implemented using official codes with necessary adaptations for single-volume scenarios. BM3D was applied slice-wise with optimized parameters. Noisier2Noise and Neighbor2Neighbor were trained on identical single-acquisition volumes and data splits as Di-Flow, also using optimized parameters based on the recommendations in the original papers. DDM$^2$ and Di-Fusion are originally designed for multi-acquisition scenarios, we adapted these methods to leverage adjacent slices within single volumes instead of corresponding slices across multiple acquisitions. Specifically, temporal redundancy was replaced with spatial redundancy by selecting the nearest $N$ adjacent slices (where $N$ matches original multi-acquisition count). Network architectures and training procedures followed original specifications with hyperparameters tuned for our datasets.

Table 1: Quantitative evaluation on 7T and 3T TOF-MRA datasets. Mean vascular SNR and CNR (with standard deviation shown below each mean) are reported for all methods. Higher values indicate better vascular signal clarity and vessel-to-background contrast.

| | 7T TOF-MRA | | 3T TOF-MRA | |
|---|---|---|---|---|
| | Vascular SNR (↑) | Vascular CNR (↑) | Vascular SNR (↑) | Vascular CNR (↑) |
| Noisy slice | $27.09_{\pm7.39}$ | $20.74_{\pm5.68}$ | $15.61_{\pm6.11}$ | $9.30_{\pm3.88}$ |
| BM3D Dabov et al. (2007) | $58.11_{\pm32.59}$ | $44.50_{\pm24.97}$ | $19.38_{\pm9.58}$ | $11.50_{\pm5.75}$ |
| NR2N Moran et al. (2020) | $45.81_{\pm18.46}$ | $34.76_{\pm14.00}$ | $19.05_{\pm8.23}$ | $11.20_{\pm5.07}$ |
| NB2NB Huang et al. (2021) | $46.45_{\pm19.70}$ | $36.69_{\pm15.63}$ | $17.41_{\pm7.18}$ | $10.16_{\pm4.32}$ |
| DDM$^2$ Xiang et al. (2023) | $29.77_{\pm10.07}$ | $22.94_{\pm8.08}$ | $15.25_{\pm5.72}$ | $9.55_{\pm3.81}$ |
| Di-Fusion Wu et al. (2025) | $41.47_{\pm16.25}$ | $30.92_{\pm12.09}$ | $18.59_{\pm8.23}$ | $10.54_{\pm4.78}$ |
| **Di-Flow (Ours)** | $\mathbf{61.30}_{\pm32.30}$ | $\mathbf{46.17}_{\pm24.41}$ | $\mathbf{22.36}_{\pm11.56}$ | $\mathbf{12.17}_{\pm6.51}$ |

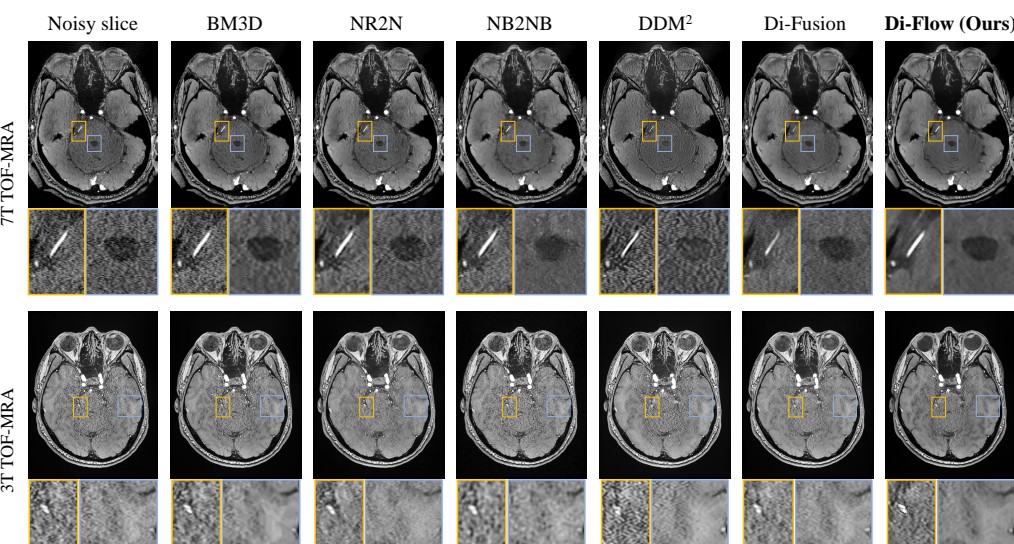

Figure 2: Visual and qualitative comparisons on 7T and 3T TOF-MRA. Each row shows denoised outputs from competing methods on representative slices. Insets highlight differences in vascular preservation and background suppression.

## 4.3 RESULTS

**Quantitative Evaluation.** Table 1 summarizes quantitative performance on 7T and 3T TOF-MRA datasets, with evaluation based on vascular signal-to-noise ratio (SNR) and contrast-to-noise ratio (CNR) metrics that respectively quantify vessel visibility and the ability to distinguish vessels from background tissue. On the 7T dataset, Di-Flow achieves the best results in both SNR and CNR, outperforming conventional, self-supervised, and diffusion-based methods. On the more challenging 3T dataset, where lower field strength yields reduced image quality, Di-Flow maintains its leading performance across all metrics. Competing methods, including recent self-supervised and diffusion-based approaches, show a pronounced drop in vessel clarity and background separation. In contrast, Di-Flow robustly suppresses noise and consistently improves vessel delineation, demonstrating its generalizability even in low-SNR regimes.

**Qualtitative Analysis.** Figure 2 presents denoised 7T and 3T TOF-MRA slices for all methods. BM3D effectively suppress background noise but frequently blur vessel structures and reduce edge definition. Noisier2Noise maintains overall contrast but can oversmooth soft tissue regions, while Neighbor2Neighbor achieves strong denoising yet sometimes obscures fine vascular detail and vessel boundaries. DDM$^2$ shows limited noise suppression in single-volume settings, and Di-Fusion

produces natural-appearing results but with diminished vessel conspicuity. In contrast, Di-Flow preserves sharp vessel edges and subtle vascular branches, balancing robust noise reduction with accurate anatomical detail. Highlighted insets confirm that Di-Flow consistently maintains small vessel visibility without introducing artifacts, supporting its quantitative advantage.

**Impact on MIP Renderings.** Figure 3 further demonstrates the clinical relevance of the proposed method using Maximum Intensity Projection (MIP) renderings. In the original noisy image, vessel structures appear fragmented, and small arterial branches are barely visualized due to background noise. After denoising, these structures exhibit noticeably improved continuity and become more visually discernible, including low-contrast peripheral vessels that were previously obscured. Our approach effectively enables clearer visualization of vascular continuity while suppressing noise and maintaining anatomical fidelity.

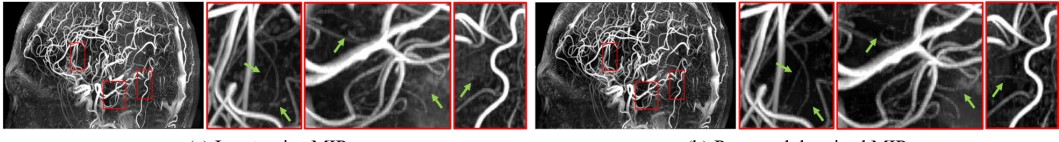

(a) Input noisy MIP                           (b) Proposed denoised MIP

Figure 3: Qualitative visualization of denoising results on 3D MRA data. (a) Original noisy MIP image, and (b) denoised output from our proposed method. The bottom row shows zoomed-in regions corresponding to the red boxes, where fine peripheral vessels that were previously obscured by noise become clearly visible in (b), as indicated by green arrows.

### 4.4 ABLATION STUDY

**Analysis on Expectation-Only Sampling.** We ablate the inference strategy by comparing expectation-only sampling with standard stochastic sampling (Figure 4). In medical imaging, determinism is critical for reproducibility and diagnostic reliability: stochastic sampling introduces run-to-run variability that can subtly alter vessel visibility. Empirically, expectation-only sampling yields more clearer restorations while enhancing vessel visibility with homogeneous background and consistently sharper tissue boundaries, whereas stochastic standard sampling exhibits background texture fluctuations and occasional spurious high-intensity artifacts (highlighted yellow boxes).

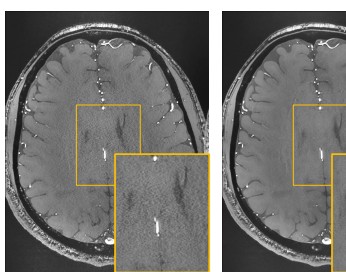

(a) Standard Sampling    (b) Expectation-Only Sampling

Figure 4: Comparison of sampling strategies.

**Analysis on PAP.** Evaluation of the PAP module revealed further improvements in local anatomical accuracy. Figure 5 illustrates that the effectiveness of the PAP module is demonstrated by comparing outputs with and without its application. Without PAP, generative denoising can lead to over-suppression or artificial generation of high-intensity vessel regions, resulting in either missing or spurious vascular structures. When PAP is incorporated, these effects are substantially mitigated. The final output maintains effective noise

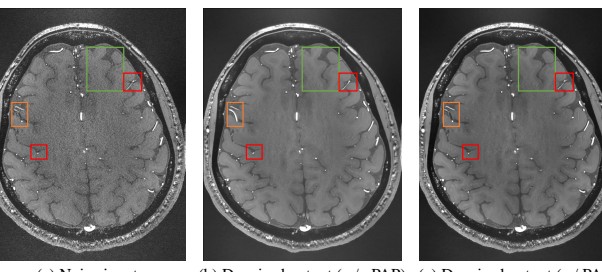

(a) Noisy input    (b) Denoised output (w/o PAP)    (c) Denoised output (w/ PAP)

Figure 5: Effect of the PAP module. Red and orange boxes indicate refined vascular details after applying PAP, while green boxes highlight regions where the denoising effect is maintained.

removal while refining only subtle local differences, particularly within vessel-rich regions. Highlighted regions in Figure 5 show that vessel boundaries become more coherent and faint vessels are preserved, confirming that PAP leverages patch statistics to restore anatomical fidelity. However, when compared to other methods, these approaches are less effective with the PAP module, as their

difference map histograms exibit instability under threshold-based clipping. This demonstrates that both the deterministic expectation-only sampling and the patch-wise adaptive correction independently and jointly enhance the quality and reliability of the final denoised output.

## 5 DISCUSSIONS

**Limitation of MSA.** The effectiveness of MSA relies on the assumption that anatomical structures exhibit sufficient coherence across adjacent slices. This assumption is well-founded for thin-slice acquisitions typical in high-resolution MRA, where vascular structures maintain consistent morphology across neighboring sections. However, in protocols with larger inter-slice gaps, the structural similarity between adjacent slices may diminish, potentially leading to suboptimal pseudo-clean target generation. In such cases, averaging could attenuate subtle slice-specific details or introduce mild blurring at structural boundaries, which may limit the generalizability of the approach.

**Dataset Generalization.** Our experiments were performed on TOF-MRA data acquired at both 3T and 7T in healthy volunteers. Across these settings, the method consistently enhanced vascular SNR and CNR, suggesting robustness to variations in field strength and acquisition quality. These results indicate promising potential for broader applicability, and further validation on more extensive, multi-vendor, and pathological datasets would strengthen evidence for its clinical utility.

## 6 CONCLUSION

We presented Di-Flow, a fully self-supervised denoising framework tailored for single noisy MR angiography volume, which operates without clean reference data, repeated acquisitions, or temporal consistency. To overcome the limitations of prior methods in low-redundancy settings, Di-Flow introduces a slice-aware pseudo-pairing strategy that leverages intra-volume structural coherence to construct effective supervision from a single volume. During inference, a conditional diffusion model is combined with expectation-only sampling and patch-wise anatomical refinement, enabling suppression of noise while preserving fine-grained structural details.

We validated our method on 7T and 3T TOF-MRA datasets. Di-Flow consistently demonstrated superior denoising performance compared to conventional, self-supervised, and diffusion-based baselines. In particular, it effectively restored weak vascular signals and enhanced anatomical continuity, especially in challenging imaging conditions such as low SNR or sparse contrast. While our experiments focused on TOF-MRA, the core framework of Di-Flow does not rely on specific anatomical priors or acquisition redundancy, suggesting its potential applicability to other volumetric modalities. These findings highlight the practical value of Di-Flow as a general-purpose denoising solution in retrospective or resource-limited clinical workflows.

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

# A $J$-INVARIANT IN DI-FLOW

In this Appendix, we provide additional technical details on the relationship between the $J$-invariant property and the Di-Flow framework. Specifically, we clarify how Di-Flow leverages $J$-invariance through the Multi-Slice Averaging (MSA) strategy and the expectation-only sampling approach, distinguishing it from prior methods such as blind-spot networks or pixel-replacement strategies.

## A.1 $J$-INVARIANT PROPERTY

The $J$-invariance concept was originally proposed in the Noise2Self framework Batson & Royer (2019) and has since been foundational to self-supervised denoising approaches. This concept is particularly valuable in clinical scenarios where paired noisy–clean datasets are unavailable, making supervised methods impractical.

**Definition.** Let $x \in \mathbb{R}^m$ be an image, and let $\mathcal{J} = \{J_1, \ldots, J_K\}$ be a partition of the pixel indices $\{1, \ldots, m\}$. A function

$$f \colon \mathbb{R}^m \to \mathbb{R}^m$$

is called $J$-invariant if, for every subset $J \in \mathcal{J}$, the output $f(x)_J$ is independent of the input values $x_J$. Formally,

$$f(x)_J \perp x_J, \qquad \forall J \in \mathcal{J}. \tag{9}$$

Intuitively, this condition ensures that predictions for pixels in $J$ depend only on pixels outside of $J$, preventing trivial identity solutions and enforcing meaningful spatial correlations.

**Proposition.** Let $f$ be $J$-invariant and assume independent pixel noise. If $f$ is an unbiased estimator of the clean image $y$, then for each $J \in \mathcal{J}$,

$$\mathbb{E}\big[\|f(x)_J - x_J\|^2\big] = \mathbb{E}\big[\|f(x)_J - y_J\|^2\big] + \mathbb{E}\big[\|y_J - x_J\|^2\big]. \tag{10}$$

Minimizing the left-hand side (self-supervised loss) thus equivalently minimizes the supervised loss plus a constant noise variance term. The optimal $J$-invariant predictor is

$$f^*(x)_J = \mathbb{E}[y_J \mid x], \tag{11}$$

where $y$ denotes the clean image and $J^c$ the complement of $J$.

## A.2 RELATION TO DI-FLOW

In prior methods, enforcing $J$-invariance typically involved explicit architectural constraints such as blind-spot networks Krull et al. (2019); Laine et al. (2019); Wu et al. (2020) or masking strategies Lehtinen et al. (2018); Batson & Royer (2019). The Di-Flow framework, however, achieves $J$-invariance implicitly by combining the Multi-Slice Averaging (MSA) strategy with expectation-only sampling during inference.

Formally, given a set of noisy slices $\{y_k\}_{k \in K}$, we construct two averaged images with target weight $w = 1$:

$$x_0 := \frac{1}{K} \sum_{k \in K} y_k = S_0 + n_0, \tag{12}$$

$$x_c := \frac{1}{K-1} \sum_{k \in K_{J^c}} y_k = S_{J^c} + n_{J^c}, \tag{13}$$

where $S_0, S_{J^c}$ are averaged clean signals and $n_0, n_{J^c}$ are averaged noise terms. Under unbiased, independent noise,

$$\mathbb{E}[x_0] = S_0, \qquad \mathbb{E}[x_c] = S_{J^c}. \tag{14}$$

During inference, Di-Flow generates two denoised estimates via expectation-only sampling: one from the full average $x_0$ and another from the contextual-only average $x_c$. The clean target signal $s_t$ is then recovered by

$$\mathbb{E}\big[s_J \mid x_{J^c}\big] = K \cdot \mathbb{E}\big[S_0 \mid x_{J^c}\big] - (K-1) \cdot \mathbb{E}\big[S_{J^c} \mid x_{J^c}\big]. \tag{15}$$

which exactly matches the optimal $J$-invariant predictor. By doing so, Di-Flow inherently enforces $J$-invariance, extracting clean anatomical structures without explicit architectural constraints or specialized loss formulations. Thus, the combination of MSA and expectation-only sampling provides a theoretical grounding for achieving optimal self-supervised denoising performance in practical clinical scenarios.

## B HYPERPARAMETER SELECTION FOR MULTI-SLICE AVERAGING (MSA)

**Rationale.** We provide detailed analysis of how MSA parameters $K$, number of slices, and $w$, target weight, were selected based on dataset characteristics and clinical requirements. Given a target index $i$ and its neighborhood $\mathcal{N}(i)$ (of size $K$), the MSA output is

$$x_i = \frac{w\, y_i + \sum_{k \in \mathcal{N}(i) \setminus \{i\}} y_k}{w + (K-1)}. \tag{16}$$

For interpretability, we also report the normalized target contribution

$$\alpha = \frac{w}{w + (K-1)} \quad \Longleftrightarrow \quad w = \frac{\alpha\,(K-1)}{1 - \alpha}. \tag{17}$$

Small $w$ (low $\alpha$) yields stronger averaging/blur; large $w$ (high $\alpha$) preserves the target slice but leaves more noise.

**Metric ROIs.** To quantify the trade-offs, we define vessel and background regions of interest (ROIs) within selected image patches. The vessel ROI captures high-intensity vascular structures, while the background ROI corresponds to the most homogeneous region (minimum variance) within the same neighborhood. We then compute:

$$\mathrm{SNR} = \frac{\mu_{\text{vessel}}}{\sigma_{\text{bg}}}, \qquad \mathrm{CNR} = \frac{\mu_{\text{vessel}} - \mu_{\text{bg}}}{\sigma_{\text{bg}}}. \tag{18}$$

**Analysis of Slice Number $K$.** We fix $\alpha$ (i.e., adjust $w$ per $K$ so that $\alpha$ stays constant) and analyze SNR/CNR vs. $K$ to isolate the benefit of adding more slices from the confound of changing the target weight. For 7T we fix $\alpha=0.60$ and sweep $K \in \{3, 5, 7, 9\}$; for 3T we fix $\alpha=0.55$ and sweep $K \in \{5, 7, 9, 11, 13\}$. This supports selecting smaller $K$ for high-SNR, thin slices (7T) and larger $K$ for lower-SNR, thicker slices (3T), while staying on the plateau to avoid unnecessary compute and $z$-axis blurring. Based on this analysis, we selected $K = 5$ for 7T and $K = 9$ for 3T, where both datasets achieve optimal SNR and CNR simultaneously.

**Analysis of Target Weight $w$.** We fix the clinically reasonable $K$ values determined above ($K=5$ for 7T, $K=9$ for 3T) and vary $w$ to examine the blur$\leftrightarrow$noise trade-off. Small $w$ provides clean background but blurs thin vessels, while large $w$ sharpens vessels but introduces visible noise speckle in low-signal regions. Through systematic evaluation, we identified $w = 6$ for both datasets as the optimal balance point where noise levels remain acceptably low while avoiding excessive blurring of fine vascular structures. This moderate weighting preserves vessel fidelity without compromising the noise suppression benefits of multi-slice averaging.

