# OpenReview forum: "Di-Flow: Self-Supervised Denoising for Single Volume MR Angiography via Slice-Aware Diffusion"
_ICLR.cc/2026/Conference — ICLR 2026 Conference Withdrawn Submission_

### Official Review · Reviewer_Xyfc · 2025-11-01

**Soundness:** 3
**Presentation:** 3
**Contribution:** 2
**Rating:** 2
**Confidence:** 4

**Summary:**

The paper proposes Di-Flow, a self-supervised denoising framework tailored for single-volume magnetic resonance angiography (MRA).
Unlike existing approaches that require clean–noisy pairs or multiple acquisitions, Di-Flow exploits intra-volume structural coherence across adjacent slices to construct pseudo-training pairs.

The model integrates three main components:

- Multi-Slice Averaging (MSA), which builds pseudo clean/noisy pairs from neighboring slices, enabling self-supervised training.

- Conditional Denoising Diffusion Model (CDDM), a two-branch diffusion framework that uses contextual priors during denoising.

- Patch-wise Adaptive Post-processing (PAP), a heuristic patch-level correction for residual artifacts.

The method is evaluated on 7T and 3T TOF-MRA datasets, showing quantitative and qualitative improvements over supervised, self-supervised, and diffusion-based baselines.

**Strengths:**

- The paper addresses a real clinical bottleneck: lack of clean labels or repeated acquisitions for supervised denoising. The single-volume setting reflects actual MRI workflows, making the work potentially deployable in clinical imaging pipelines.
- Leveraging intra-volume redundancy is a thoughtful idea, bridging a gap between single-image self-supervision and multi-volume generative methods.
- Demonstrates solid quantitative gains (SNR and CNR) across field strengths. Qualitative visualizations show convincing improvements in small-vessel visibility, which is critical for angiography. Ablation studies (expectation-only vs stochastic; with vs without PAP) provide evidence that each component contributes measurable benefit.
- The writing is clear and easy to follow.

**Weaknesses:**

- The auxiliary self-reconstruction path (Section 3.2) is under-explained. It’s unclear how this branch works. Please provide a mathematical justification (e.g., show how joint optimization encourages feature disentanglement or anatomical prior learning), or include an ablation removing the self-branch to quantify its impact.
- Section 3.2.1 presents expectation-only inference as a core methodological component, but this is a standard deterministic sampling technique already used in diffusion-based restoration (e.g. DDIM). The authors could move this detail to the implementation or inference appendix rather than framing it as a standalone innovation.
- PAP (Section 3.3) appears manually engineered and dataset-specific, relying on hand-tuned thresholds and patch classification rules. While the intuition of histogram-based correction is reasonable, this component undermines the otherwise clean learning-based framework. In addition, it’s unclear whether PAP can generalize to other datasets or modalities.

**Questions:**

- How exactly is the self-reconstruction branch trained? Does it share weights with the primary denoising branch? How are the losses from the two branches combined? Does the self-reconstruction branch influence inference at all?
- How sensitive is the J-invariant assumption to non-independent noise across slices (a realistic scenario in MRI acquisition)?
- The PAP module relies on manually defined thresholds and patch classification rules. How are these thresholds chosen? Are they dataset-specific, or fixed across 3T and 7T data?

---

### Official Review · Reviewer_cb9h · 2025-11-01

**Soundness:** 2
**Presentation:** 3
**Contribution:** 2
**Rating:** 2
**Confidence:** 4

**Summary:**

In this paper, the authors propose a self-supervised denoising framework for single 3D medical image volumes, with a focus on magnetic resonance angiography (MRA). The key idea is to exploit spatial coherence between adjacent slices to form training pairs, thus avoiding the need for clean ground truth or multiple scans. The core denoising engine is a conditional diffusion model with expectation-only sampling, which aims to recover the underlying signal in a deterministic fashion. To further enhance anatomical fidelity, the authors introduce a patch-wise adaptive post-processing (PAP) module to refine local image structures. Experiments on 7T and 3T time-of-flight in-vivo MRA datasets demonstrate improved vessel visibility and reduced noise compared to baseline approaches.

**Strengths:**

1. The paper tackles a practically important problem: self-supervised denoising for single 3D medical volumes, where clean data are unavailable..
2. The use of adjacent slices for training pair construction is conceptually sound and well-motivated.
3. The results, both qualitative and quantitative, appear visually convincing and indicate potential clinical value.

**Weaknesses:**

Major concerns:
1. The so-called “patch-wise adaptive post-processing” is essentially a soft-thresholding operation, which raises questions about novelty and necessity. The need for this correction suggests the diffusion model may be missing a residual skip connection or is miscalibrated in intensity. The authors should verify whether the residual output has zero mean, as this could eliminate the need for manual intensity correction. Importantly, this step is crucial yet not properly ablated. A quantitative table showing results with vs. without PAP for all studied methods would be essential to assess its impact. Improving these ablations would strongly clarify the contribution, and I'd gladly increase my score when this point is clarified.
2. More generally, the ablation studies are insufficient. Showing a few visual samples cannot substitute for numerical analysis. The authors should provide quantitative metrics (even relative ones) to support their design choices.
3. Reference-free image quality metric is a rich and vast research domain, but the metrics chosen by the authors are unknown to the community. The authors should at least add other another metric (e.g. NIQE or MUSIQ).

Minor concerns:
1. Equation (4) is presented without sufficient context or justification. The proposed “expectation-only sampling” is reminiscent of deterministic diffusion or flow-matching approaches, yet the connection is never discussed. The authors should explain how their formulation relates to recent methods, and whether similar non-stochastic sampling schemes have been proposed before.
2. Line 192: I think that the interpretation of $\alpha$ and $1-\alpha$ is inverted: $1-\alpha$ controls the cumulative noise strength, meaning that larger $\alpha$, corresponds to less noise (and not the opposite).
3. Section 3.2.2 (misnamed “J-invariant”) does not clearly justify the use of the J-invariance principle. Conceptually, J-invariance is less about enforcing an optimal predictor and more about constructing IID noisy samples of the same image to mimic a noise2noise setting. The current justification in appendix is not satisfying and could be removed without removing interest to the paper. Lastly as currently written, section 3.2.2 could be merged into the main text; it does not warrant a separate heading.

**Questions:**

1. My main concern is that of PAP relevance in general and for other methods as well. Showing that both the proposed deep learning based module outperforms other modules + that PAP is beneficial for all modules would be a strong addition to the paper which would make me increase my score.
2. Could the authors confirm that no residual skip connection is missing in their architecture and that the restored patch residual is indeed 0 mean?
3. Did the authors try wavelet based soft-thresholding insead of image thresholding?

---

### Official Review · Reviewer_ZQca · 2025-11-01

**Soundness:** 2
**Presentation:** 2
**Contribution:** 2
**Rating:** 2
**Confidence:** 4

**Summary:**

This paper proposes a diffusion-model-based framework for self-supervised denoising in MR angiography. The method reformulates the diffusion model input as two averaged adjacent MR slices, denoted as the pseudo-clean and contextual prior images, with the latter serving as conditional input. After analytically reverting the predicted averages back into slices, the authors introduce heuristic post-processing strategies to further refine residual noise. The use of averaged slices as denoising targets and the subsequent post-processing scheme are original. However, the work suffers from insufficient ablation studies, lack of hyperparameter sensitivity analysis, and limited justification for key design choices. Concerns also exist regarding metric computation and consistency in argumentation. Overall, substantial revision is required before this work is suitable for publication.

**Strengths:**

1.	The concept of using averaged adjacent slices as self-supervised denoising targets in MR angiography is original.
2.	The post-processing module is carefully designed and novel.
3.	The proposed method achieves clearly improved vascular SNR/CNR over previous approaches.

**Weaknesses:**

1. Metric reliability: The definition of the vascular ROI for SNR/CNR computation is unclear. Since real 7T/3T datasets are treated as noisy scans, it is questionable how “clean” ROIs are obtained. If Eq. (6)’s vessel-dominant pseudo-class is used, post-processing may leverage information related to the evaluation metrics, compromising fairness.
2. Lack of ablation studies: Although the method improves vascular SNR/CNR, no quantitative analysis isolates the contributions of each proposed component.
3. Unspecified hyperparameters: The patch-wise adaptive post-processing lacks parameter details and tuning rationale.
4. Missing sensitivity analysis: The framework introduces multiple new hyperparameters without analyzing their sensitivity or robustness.
5. Limited rationale for averaging: The choice of averaged adjacent slices is weakly justified; as noted in Section 5, larger inter-slice gaps may invalidate this assumption. Visual comparisons among x_0, x_c, and target slices would help clarify effectiveness.
6. Missing runtime evaluation: Inference time comparisons with baseline methods are omitted.
7. Overstated “single-volume” claim: The method trains on multiple patients but claims single-volume denoising, without patient-specific results to support this.
8. Unclear guidance operation: In Figure 1 (training phase), the “guidance” operation for x_cis confusing and lacks corresponding equations.
9. Equation inconsistency: In Eq. (2), subscripts/superscripts should clearly distinguish time steps (0–T) and image types (x_0, x_c) for clarity.
Therefore, the experimental design and results exhibit several issues, which make the paper unsuitable for acceptance in its current form.

**Questions:**

Questions
1.	What are the quantitative contributions of each proposed component?
2.	What are the results when applying the proposed post-processing to previous methods?
3.	How sensitive is the performance to new hyperparameters in multi-slice averaging and post-processing? (For fairness, previous methods should also be tuned accordingly.)
4.	How is the vascular ROI defined for SNR/CNR computation? Does post-processing access ROI-related information?
5.	What are the inference times across compared methods?
6.	Why use DDPM reverse sampling without noise instead of DDIM?
7.	How does the framework perform in patient-specific (“single-volume”) reconstruction? In this setting, what are the training/inference costs compared to other methods?

---

### Official Review · Reviewer_hVh3 · 2025-11-04

**Soundness:** 3
**Presentation:** 4
**Contribution:** 3
**Rating:** 6
**Confidence:** 3

**Summary:**

This paper presents Di‑Flow, a self‑supervised denoising framework tailored for single‑volume MR angiography. The approach is intentionally simple and effective: it constructs pseudo training pairs from adjacent slices, learns with a conditional diffusion model, and performs deterministic inference. The method is evaluated on 7T and 3T MR datasets and demonstrates clear state‑of‑the‑art denoising performance for the single‑volume setting.

**Strengths:**

- Clear and effective methodology. The overall pipeline is easy to follow and well motivated for single‑volume data.

- Thorough experiments on 7T and 3T datasets. The experimental design and analyses are meaningful, and the accompanying discussions are insightful.

- Strong results. The method achieves state‑of‑the‑art performance for single‑volume MR denoising, with visibly improved vascular clarity and noise suppression.

- Great presentation. The paper is clearly written, with a smooth narrative flow and helpful figures.

**Weaknesses:**

- As also noted by the authors, baselines such as DDM^2 and Di‑Fusion were originally designed for multi‑volume or multi‑acquisition data (i.e., they depend on slice‑wise or temporal consistency). As a result, they may not transfer naturally to the single‑volume setting. For a fairer assessment of state‑of‑the‑art performance, it would help to also evaluate in multi‑volume acquisition scenarios—while keeping the proposed method’s inference per‑volume—to show how Di‑Flow compares when competing methods operate in their natural regime. Clarifying training data usage and any adaptations made to these baselines would also strengthen the comparison.

**Questions:**

- Does training Di‑Flow require multiple training volumes in addition to the target volume to be denoised? If so, the comparison with methods such as DDM^2, which can be trained using only the target volume(s), may not be strictly apples to apples. A clear statement of training‑time data requirements across methods would help interpret the results and their clinical practicality.

---

### Note · Authors · 2025-11-21

I have read and agree with the venue's withdrawal policy on behalf of myself and my co-authors.